# Five Lessons for Effectively Transitioning Problem-Based Learning to Online Delivery

Mandar Jadhav [1],[†], Deepika Shaligram [2],[†], Bettina Bernstein [3],[†], Sandra DeJong [4],[†], Jeffrey Hunt [5],[†], Say How Ong [6],[†], Anthony Guerrero [7],[†] and Norbert Skokauskas [8],[*],[†]

1 Committee on Health, Education, Labor and Pensions, United States Senate, Washington, DC 20002, USA
2 Department of Psychiatry, Boston Children's Hospital/Harvard Medical School, Boston, MA 02453, USA
3 Department of Psychiatric Medicine, Philadelphia College of Osteopathic Medicine, Philadelphia, PA 19131, USA
4 Department of Psychiatry, Cambridge Health Alliance/Harvard Medical School, Cambridge, MA 02139, USA
5 Department of Psychiatry and Human Behavior, The Warren Alpert Medical School, Brown University, Providence, RI 02915, USA
6 Department of Child and Adolescent Psychiatry, Institute of Mental Health, Singapore 539747, Singapore
7 Department of Psychiatry, University of Hawaii, Honolulu, HI 96813, USA
8 Department of Mental Health, Norwegian University of Science and Technology, 7491 Trondheim, Norway
* Correspondence: norbert.skokauskas@ntnu.no
† These authors contributed equally to this work.

**Abstract:** Problem-based learning (PBL) is an active learning technique that promotes a life-long learning approach to understanding and using the principles of clinical medicine. It does so by helping learners hone their critical thinking skills in a team-based environment. It was originally developed for use in live, in-person settings. During the COVID-19 pandemic, it has had to be rapidly adapted for online delivery. In this article, we first highlight the key challenges faced by educators and learners in making this transition. We then share five lessons for effectively translating in-person PBL curricula to online and hybrid learning formats.

**Keywords:** problem-based learning; hybrid; virtual; online; transition; COVID-19

## 1. Introduction

Problem-based learning (PBL) was popularized by Barrows and Tamblyn in the 1960s to promote a team-based, critical thinking approach to teaching clinical medicine [1]. Now, over 70% of North American medical schools incorporate PBL into their curricula [2]. At the core of PBL are cases that help learners integrate ethical, biological, and social aspects of healthcare by generating situational interest, which enhances self-directed learning and collaborative problem-solving [3]. As an active mentor-guided peer-driven learning technique, PBL engenders a more persistent understanding of the underlying principles of medicine, leading to a proclivity for life-long learning [4]. PBL is uniquely suited to prepare learners for a multipronged clinical systems approach, one that is integrative, inclusive of the biopsychosocial model, and interdisciplinary. It teaches teamwork in case formulation and treatment planning, which learners can apply effectively when interacting with other staff, patients, and caregivers [5]. Compared to traditional learning techniques, PBL is shown to be effective in improving student and faculty satisfaction with the medical learning process, and in improving clinical skills in many fields of medicine [6,7].

PBL does require more formal training of the tutor to prepare effective curricula, and requires students to have mastered basic problem-solving skills ahead of time. PBL groups typically have five to ten students and one tutor—often a clinician-educator. Groups typically meet for two or more sessions per case, during which they identify significant problems and research pertinent literature to develop solutions [8]. At the end of each case, groups re-evaluate their initial clinical reasoning [9]. Originally, this design was intended

for use in-person. However, during the COVID-19 pandemic, PBL had to be adapted for online delivery. This transition to remote learning resulted in major challenges for educators and learners.

## 2. Key Challenges Faced in Transitioning to Online PBL

1.  In-person groups rely on the seamless integration of non-verbal communication, such as making and breaking eye-contact fluidly, to direct the discussion and enable the rapid exchange of ideas [10]. It can be difficult to simulate such an environment using online tools, for instance, due to eyeline mismatch created by positioning cameras above screens.
2.  Flexibility in shifting focus in-person is often achieved by forming and dissolving mini groups transiently. This relies on being able to conduct multiple conversations simultaneously in a shared space and selectively highlighting learners' voices, which is something not all online platforms support [11].
3.  Visual tools, such as mechanistic case diagramming, rely on the use of whiteboards or paper that can fit into the learning space without competing for attention [12]. Online platforms often use the whole screen for whiteboarding.
4.  The tutor's central position during in-person PBL guides learners' attention during introductory and transitional phases. Additionally, the tutor can provide one-on-one support to learners who need help without impacting the group [10,11]. Online platforms' capabilities for facilitating joint attention can be rudimentary, while features such as the chat can be distracting.
5.  Comparative analyses of online collaboration platforms' capabilities have been limited, and systematic studies have only started to emerge in the last one to two years [13]. Thus, educators have had to rely on informal or non-scientific sources to select the right tools for their curricula and teaching environments.

## 3. Five Lessons to Ensure a Successful Transition

The American Academy of Child and Adolescent Psychiatry (AACAP) PBL Special Interest Study Group (SISG), to which the authors belong, collaborated on identifying the following five solutions for successfully transitioning PBL to an online platform.

1.  Using webcams with automatic person-tracking, along with standing desks, can improve participants' ability to share non-verbal gestures. These devices could be offered for use at home or incorporated into collaborative spaces on campus. The institutional technology team should be included early in the planning of the deployment of these devices [14].
2.  Selecting the right online platform can overcome challenges regarding participant positioning and eye contact. For instance, Microsoft Teams and Zoom offer Together Mode and Immersive View, respectively, which use artificial intelligence (AI) to position participants in stepped intervals in a shared virtual space, rather than in a flat grid of rectangles [15]. This technique enables natural focus-shifting between participants. Devices such as Microsoft Surface, Facebook Portal, and Apple iPads offer AI-powered gaze-correction. This makes it seem as if participants are looking at the camera instead of their screens, facilitating natural eye contact [16].
3.  Mini groups can be approximated using breakout rooms, a feature popularized by Zoom. This does require modifying the curriculum to prescribe time-limited mini discussions, followed by reporting learning back to the larger group. The tutor must actively manage breakout rooms lest learners leave prematurely or remain sequestered longer than necessary. Further, the chat function could be used for the facilitation of mini groups, and individual chats could be used for the one-on-one support of learners. Voice modulation for emphasis has not yet been fully realized despite the use of echo cancellation and alternate audio streams online. Learners' spaces can instead be acoustically optimized. Using the Raise Hand tool during live collaboration and

supplementing it with asynchronous online bulletin board discussions are additional workarounds [17].

4. Some platforms, such as Zoom, offer inventive viewports which can be used to superimpose participants in front of a virtual whiteboard. This allows participants to pay attention to both simultaneously. If this is not feasible, participants may be asked to use online multi-party authoring on a Learning Management System (LMS) instead. LMS providers, such as Office 365 and Google Workspace, include real-time change-tracking, which, when used on large screens or multi-monitor setups, do not prevent participants from shifting attention between people and tools rapidly [14,15].

5. Investing in faculty training in participant management tools such as Zoom's Highlight Active Speaker, Mute Participant, Remote Support, Share Screen, Waiting Room, Passcodes, and equivalent functions on other platforms, could reduce the likelihood of session-breaking interruptions. Providing learners or tutors with technical support once a group session has started is impractical and likely to result in greater loss of productivity versus training them pre-emptively [17].

Additional solutions are included in Table 1.

**Table 1.** Additional solutions for adapting PBL to online or hybrid learning.

| Challenges | Solutions |
|---|---|
| Learner engagement and participation is lower in distance or hybrid learning PBL. | • Promote active learning by integrating polls, breakout rooms, chat boxes, smartboards, and quiz show-like sessions.<br>• Set ground rules for technology use ahead of time, such as learners taking turns, and keeping cameras and microphones on.<br>• Schedule in-person, synchronous sessions. |
| Integration of learning over multiple sessions beyond one-at-a-time presentations. | • Use visual collaborative learning aids such as annotating shared screens or virtual whiteboards.<br>• Include teaching techniques such as learner recaps of previous sessions.<br>• Develop and share a course curriculum map with learners ahead of time. |
| Tutor uncertain of learner response due to reduced throughput of non-facial nonverbal expressions. | • Supplement communication with group or individual tutor–learner chat.<br>• Schedule individual follow-up sessions. |
| Potential for tutor burnout. | • Schedule more frequent, shorter breaks.<br>• Incorporate asynchronous teaching.<br>• Use "Together Mode", a virtual background. |
| Learners requiring more content and technological support. | • Have backup curriculum for unexpected interruptions in connectivity or platforms.<br>• Plan orientation to distance learning, especially for nontraditional learners.<br>• Become familiar with online learning support resources at your institution.<br>• Manage workspace to limit distractions, improve lighting, and ergonomic comfort.<br>• Emphasize facial expressions and hand/arm gestures.<br>• Enunciate clearly with more pauses and slower speech.<br>• Use a live-transcription feature. |

## 4. Future Directions

In summary, online PBL provides many of the benefits of in-person PBL, such as a more intellectually challenging, motivating, and enjoyable approach to education with the bonus flexibility of asynchronous and distance learning for learners and tutors. The use of online teaching in PBL curricula for pre-clinical undergraduate medical trainees can reinforce the use of technology in real-world clinical situations and increase the quality of learning for graduate trainees [18]. PBL tutors may also derive benefits in their own lifelong learning process by becoming more adept at using technology in distance or hybrid learning settings. Given the detrimental effects of the pandemic on learners in terms of decreased social interaction, psychological distress, and the resulting academic consequences, online PBL may provide solutions for increased access to learning for those with geographic, health, or schedule-related limitations. The use of distance and hybrid learning PBL can help improve the participation and retention for learners with different learning styles or who might be more reticent learners in in-person settings. Online and hybrid learning incorporating PBL can be used to provide more active learning opportunities, including the incorporation of multimedia, written communication, transcripts, screen captures, and other video-specific techniques [17].

We recognize that some of the solutions we highlighted may be difficult to implement due to cost or regulatory reasons [19]. A strong argument for investing in these solutions is that in places such as Singapore and New Zealand, where the COVID-19 pandemic did not hinder in-person learning to the same degree as in the United States, these solutions were used to expand the learner and educator base via hybrid PBL (see Table 2 for more details on hybrid PBL) [17]. Online delivery permitted access to subspecialty experts in geographic areas they would otherwise never have visited, and accelerated the availability of such expertise in places where training experts locally would take decades. Combined with the cultural awareness of local faculty, hybrid PBL has become a particularly attractive teaching tool globally. This shift has necessitated innovation in course structure and more investment in online platform technologies that will be useful for more than just PBL-based learning [20]. We anticipate that the global adoption of hybrid PBL will accelerate further after the pandemic to address the ongoing need for high-quality, equitable, and cost-effective medical education in distant locations. This need has been driving efforts for more than ten years in Australia, Norway, and Hawaii to make hybrid PBL successful [21]. Thus, while some platform-specific capabilities may be novel, the fundamentals of these solutions are time-tested. In a world where we must increase the diversity of voices in medical education, we anticipate that these strategies will prove critical [17]. We will continue to develop, refine, and disseminate such advances in the use of PBL in medical education at future AACAP PBL SISG meetings, and in scientific writing.

**Table 2.** Comparing features of in-person, online, and hybrid PBL.

| | In-Person PBL | Online PBL | Hybrid PBL |
|---|---|---|---|
| Structure | • Physical classroom around a table with/without a whiteboard. | • Online conferencing with/without a Learning Management System (e.g., Blackboard). | • Classroom plus online conferencing with/without an LMS. |
| Delivery Mode | • Synchronous group of 5–10 learners per tutor. | • Synchronous group of >10 learners per tutor possible.<br>• Asynchronous recorded material for individual learning.<br>• Individual online sessions at set intervals with tutor. | In addition to pre-requisites of online learning:<br>• Synchronous group of 5–10 learners per tutor online or in-person. |

**Table 2.** *Cont.*

| | In-Person PBL | Online PBL | Hybrid PBL |
|---|---|---|---|
| Teaching Tools | • In-person polling, quizzes, whiteboards, and printed articles. | • Online polling, quizzes, electronic whiteboards, searching of databases, and watching of synchronous media. | • In-person tools supplemented by online ones per tutor preference and learner group size. |
| Tutor's Role | • Is a content expert or expert in guiding PBL.<br>• Avoids unidirectional lectures to foster active learning.<br>• Provides just-in-time scaffolding.<br>• Must prepare printed material ahead of time. | In addition to pre-requisites of in-person learning:<br>• Unidirectional lectures expected for asynchronous content.<br>• May need technical support to address connectivity and scheduling issues.<br>• Must prepare workspace for camera and microphone use. | In addition to pre-requisites of online learning:<br>• Must prepare printed material, and camera and microphone use. |
| Learner's Role | • Synthesizes information.<br>• Attends on time.<br>• Uses verbal and non-verbal expression to communicate. | In addition to pre-requisites of in-person learning:<br>• Uses camera, microphone, chat, bulletin boards, and shared documents to communicate.<br>• Must limit environmental distractions. | In addition to pre-requisites of in-person and online learning:<br>• Must keep track of session format per schedule. |
| Advantages | • Collaboration more engaging in-person.<br>• Many-to-many and smaller cohort concurrent discussions can occur with fluid interruptions.<br>• Teamwork skills translate more directly to clinical settings. | • Integration of different learning styles.<br>• Turn-based discussions allow for more uniform participation.<br>• Cohort discussions are possible using "breakout rooms".<br>• More learners and geographic access.<br>• Accommodates different schedules.<br>• Accommodates learner or tutor health concerns. | • Allows for engaging in-person collaboration and integration of learning styles.<br>• Fluid discussions possible during in-person sessions<br>• Enables increased access to learning.<br>• Teamwork skills translate more directly to clinical settings. |

**Author Contributions:** M.J., D.S., B.B., S.D., J.H., S.H.O., A.G. and N.S. were all involved in the conceptualization and writing, including original draft preparation and the review and editing of this manuscript. All authors have read and agreed to the published version of the manuscript.

**Funding:** This research received no external funding.

**Institutional Review Board Statement:** Not applicable.

**Informed Consent Statement:** Not applicable.

**Data Availability Statement:** No new data were created or analyzed in this study. Data sharing is not applicable to this article.

**Conflicts of Interest:** The authors declare no conflict of interest.

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
