# Peer review of "Five Lessons for Effectively Transitioning Problem-Based Learning to Online Delivery"

_ime, doi:10.3390/ime2010004_

Round 1
Reviewer 1 Report
This commentary addresses an important topic in a timely manner. However, because of the large number of papers on this topic, it may be difficult to capture the reader’s interest. I add some suggestions which may help to provide a more focussed view.
- the authors quote differences in the several platforms for online PBL: they should provide a table highlighting the differences with more technical details and practical solutions;
- table 2 is rather difficult to read because of the many repetitions in the different columns, which do not allow to capture differences and advantages of the different formats;
- self-citations should be limited to a few references (in the present version, one third of the citations falls within this category);
- does the paper provide “five lessons” (title) or “five ways” (text): answering this question might help to make the message more straightforward to the reader.
Reviewer 2 Report
The authors have given a well versed perspective communication manuscript on the PBL modes and discrepencies.
Author Response
Thank you for your feedback.
Reviewer 3 Report
This paper carefully discusses the problems and solutions, which would be interesting for faculty member at medical school. This paper is publishable with minor revisions in my opinion.
#1 Five ways to ensure a successful transition
There is no explicit solution in the section of "Five ways to ensure a successful transition" for the key challenges that "the tutor's central position during in-person PBL guides.......(L53-)". For example, a chat function could be used for facilitation, or individual chats could be used for one-on-one support.
#2 Future Directions
In the text, it would be better to describe some of the most important benefits of online PBL from Table2.
Reviewer 4 Report
1. The article may benefit from a longer discussion of the current barriers and enablers of PBL. While cost or regulatory reasons are listed as current challenges that may inhibit its development, the authors should better capture the problem for the benefit of readership.
2. In addition to the previous point, the article can benefit from a more explicit representation of which developments/improvements will support the adoption of PBL globally. These could be in terms of technological innovations, course structure or investment driven.
3. Along with the disruptions in face-to-face teaching due to COVID-19, the authors may want to show that this has had detrimental effects for learners in terms of social interaction, decreased motivation, or aggravated mental health and other psychosocial challenges (doi.org/10.1016/j.dsx.2020.05.035, doi.org/10.3390/su14159699, doi.org/10.1016/j.hlc.2020.05.002).
4. Point 3 could also be discussed in the context of PBL. How will the important social benefits of in-person learning be safeguarded in a future scenario of wider PBL adoption?
Round 2
Reviewer 4 Report
Authors have fulfilled most of my concerns, however have disregarded the acknowledgement of the proposed references under my previous comment (point 3). I believe their inclusion would augment the soundness of what's discussed in that particular case